# Long-Term Satisfaction and Patient-Centered Outcomes of Deep Brain Stimulation in Parkinson’s Disease

**DOI:** 10.3390/brainsci8040060

**Published:** 2018-04-01

**Authors:** Jessica A. Karl, Bichun Ouyang, Kalea Colletta, Leo Verhagen Metman

**Affiliations:** 1Movement Disorder Section of Neurological Sciences, Rush University Medical Center, 1725 W. Harrison Street, Suite 755; Chicago, IL 60612, USA; bichun_ouyang@rush.edu (B.O.); lverhage@rush.edu (L.V.M.); 2Department of Neurological Sciences, Loyola University Medical Center, 2160 S. 1st Ave.; Maywood, IL 60153, USA; kcolle2@gmail.com

**Keywords:** deep brain stimulation, Parkinson’s disease, subthalamic nucleus, patient-centered outcomes, quality of life, satisfaction, long term

## Abstract

Bilateral subthalamic nucleus (STN) deep brain stimulation (DBS) is an effective and proven treatment option for patients with advanced Parkinson’s disease (PD). Long-term outcomes (>5 years) have demonstrated sustained improvement in objective motor symptoms; however, few studies have evaluated patient-centered outcomes other than quality of life (QOL). A locally developed DBS-patient-centered outcomes questionnaire was administered to PD patients >5 years post-DBS. All questions were scored on a ten-point scale, whereby 0 represented the most ‘positive’ answer and 10 the most ‘negative’ answer. Pre-operative scales were repeated at the time of survey. Fifty-two patients (mean 8.2 ± 2.6 years post-DBS) were included. Satisfaction was high with median score (range) of 1/10 (0–8) at the time of survey. Patients endorsed having made the correct decision by undergoing DBS, with a score of 0 (0–10), would choose to have DBS again, with a score of 0 (0–10), and would recommend DBS to others, with a score of 0 (0–10). Pre-operative expectation target was set at a high level with a score of 2 (0–10). Parkinson’s Disease QOL (PDQ-39) Questionnaire Summary Index (SI) scores were, mean (SD), 2.1 (18.2) above baseline (*p* = 0.44). Those with worsening in PDQ-39-SI scores had less satisfaction with DBS (*r_s_* = 0.57, *p* ≤ 0.0001). This is the first study to assess long-term patient satisfaction with STN DBS. We are currently collecting data prospectively to confirm the results of these preliminary findings.

## 1. Introduction

Bilateral subthalamic nucleus (STN) deep brain stimulation (DBS) is an effective and proven treatment option for patients with advanced Parkinson’s disease (PD). Long-term outcomes (>5 years) have demonstrated sustained improvement in objective motor symptoms [1,2,3]. There have been few studies evaluating patient-centered outcomes other than quality of life (QOL). Health-related QOL appears to be variably maintained and can differ between individual domains [4,5]. Hasegawa et al. [6] did evaluate patients’ perspectives of DBS outcomes in the short term (1 year) and found no correlation between Parkinson’s Disease QOL (PDQ-39) Questionnaire scores and overall satisfaction, but did find that preoperative counseling of patients’ expectations did positively affect patient satisfaction. Unrealistic expectations of DBS surgery can lead to reduced satisfaction amongst patients and caregivers, even in the face of ‘objective’ motor improvement. There has been no exploration of satisfaction and expectations in the long term (>5 years) and overall long-term patient-centered outcomes of DBS are lacking. The Patient-Centered Outcomes Research Institute (PCORI) stipulates a goal of, “…evaluating questions and outcomes meaningful and important to patients and caregivers [7].” Accordingly, in this study we determined long-term patient satisfaction, QOL, and self-reported disability after STN DBS. In addition, predictors of satisfaction and the relationship between satisfaction and changes in QOL and symptom severity were investigated. 

## 2. Methods

PD patients who received bilateral STN DBS at Rush University Medical Center (RUMC) a minimum of five years earlier were identified. The study was approved by the Rush University Institutional Review Board (ID: 14082201; Date of approval 15 October 2014). Each participant provided informed consent. At the time of surgery, extensive multi-disciplinary testing had been completed by the RUMC DBS team to determine patients’ eligibility for DBS surgery, including evaluations by a movement disorder neurologist, neuropsychologist and neurosurgeon. Standard rating scales, including Unified Parkinson’s Disease Rating Scale (UPDRS) I–IV, Schwab and England scale (S&E), PDQ-39, as well as pre-operative demographics and Levodopa equivalent daily dose (LEDD) were available for review in all cases. After study entry, current demographics were obtained and the PDQ-39, UPDRS-I, UPDRS-II, S&E and UPDRS-IV were repeated to evaluate changes in QOL, mood and cognition, disability, and complications of therapy. In addition, current LEDD was calculated.

### Patient-Centered Outcomes

To evaluate post-operative satisfaction, a locally developed DBS patient-centered outcomes questionnaire (RUSH-DBS-Q) was administered (Appendix A). The RUSH-DBS-Q is divided into two sections. The first section focuses on patients’ pre-operative expectations, their theoretical decision to undergo DBS again, their sense of timing of DBS, their confidence in recommending DBS to others, and their overall satisfaction with DBS. The second section asks patients to rate the severity of their motor (dyskinesia, motor fluctuations, tremor, rigidity, bradykinesia, gait, balance, dystonia) and non-motor (cognitive impairment, depression, apathy, insomnia, excessive daytime sleepiness, pain, PD medication side effects) symptoms before surgery and at the current time. All questions are scored on a ten-point scale, whereby 0 represents the most ‘positive’ answer and 10 the most ‘negative’ answer. Patients and their caregivers filled out the RUSH-DBS-Q an average of eight years post-DBS surgery, so recall and other bias is inherent. However, a substantial portion of the questionnaire targeting post-operative outcomes was completed at the appropriate time. 

The questionnaires were mailed to patients prior to the interview to give patients and caregivers the chance to familiarize themselves with the content, and once they indicated to be prepared the interview was conducted. Data were collected in the outpatient clinic or over the phone according to patient/caregiver preference. Patients and caregivers worked together to answer all questions. 

## 3. Statistical Analysis

Descriptive analysis (mean, median, standard deviation, range) was obtained for each variable. To evaluate predictors of outcome, the pre-operative UPDRS-III sub scores were grouped into the following factors: axial (items 29–30), resting tremor (item 20), postural tremor (item 21), bradykinesia, (items 23–26) and rigidity (item 22) [8]. Comparison of clinical data pre-operatively and at time of study was completed using the paired t-test for normally distributed data and the Wilcoxon signed rank test for non-parametric data (statistical significance determined at *p* < 0.05). 

Patient ratings on the RUSH-DBS-Q of motor and non-motor symptoms before surgery and at the current time were compared. If scores were identical these symptoms were labeled “no change”, if scores were lower at the current time the symptoms were labeled “better” and if scores were higher those symptoms were labeled “worse”. 

Spearman correlation analysis was completed to evaluate the relationship between post-operative satisfaction scores and reported symptom severity, pre-operative expectations, and pre-operative response to levodopa (change in baseline UPDRS-III-OFF to baseline UPDRS-III-ON). The relationship between change in LEDD from pre-DBS to current time and apathy was also investigated. 

## 4. Results

### 4.1. Participants

Ninety-four PD patients were identified who received simultaneous bilateral STN DBS a minimum of five years earlier. Fifty-two patients (thirty-one male, twenty-one female) were enrolled, thirty-one were lost to follow up of whom at least twelve were deceased, six were non-English-speaking, five refused to participate, and one had been explanted. Patients who received unilateral STN or DBS in other targets were excluded from participation.

### 4.2. Rating Scales and LEDD

Demographic and baseline pre-operative data are described in Table 1. After 8.2 ± 2.6 years, there was no change in the UPDRS-II-OFF score (*p* = 0.66), S&E-OFF score (*p* = 0.71) and the PDQ-39 summary index (SI) scores (*p* = 0.44), and sustained improvement in the UPDRS-IV score (*p* ≤ 0.0001). LEDD was considerably lower (*p* < 0.0001). Conversely, worsening was seen in the UPDRS-1 score (*p* = 0.0002), S&E-ON score (*p* < 0.0001), and UPDRS-II-ON score (*p* < 0.0001) (Table 2).

### 4.3. Rush Patient-Centered Outcomes Questionnaire (RUSH-DBS-Q)

Satisfaction was rated high by the majority with 65% of patients scoring a 0, 1 or 2 evidenced by a median score (range) of 1 (0–8). Patients endorsed having made the correct decision by undergoing DBS, with a score of 0 (0–10), would choose to have DBS again, with a score of 0 (0–10), and would recommend DBS to others, with a score of 0 (0–10). The majority of patients indicated that they would have preferred surgery at an earlier time with 63.5% of patients scoring a 0, 1 or 2, leading to a median score of 1 (0–10). (Figure 1). 

Worsening in patient-reported outcomes from pre-DBS to current time (8.2 ± 2.6 years) as measured by the PDQ-39 Summary Index (*r_s_* = 0.57, *p* ≤ 0.0001), PDQ-39 mobility domain (*r_s_* = 0.57, *p* ≤ 0.0001), PDQ-39 emotional well-being domain (*r_s_* = 0.40, *p* = 0.005), PDQ-39 stigma domain (*r_s_* = 0.40, *p* = 0.005), PDQ-39 social support domain (*r_s_* = 0.45, *p* = 0.001), and S&E-OFF (*r_s_* = −0.44, *p* = 0.002) correlated with lower satisfaction. There was no association between pre-operative levodopa responsiveness and satisfaction (*r_s_* = −0.01, *p* = 0.93) (Table 3). There were no predictors of post-operative satisfaction found (*p* = 0.95) (Appendix A).

#### 4.3.1. Expectations

The pre-operative expectation target was set at a high level with 53% of patients scoring a 0, 1 or 2 evidenced by a median score (range) of 2 (0–10). The correlation between pre-surgical expectations and current satisfaction was not significant (*r_s_* = 0.27, *p* = 0.06). 

#### 4.3.2. Patient-Rated Motor and Non-Motor Symptoms

There was a weak but significant correlation between current satisfaction and patient-rated severity in motor (*r_s_* = 0.36, *p* = 0.01) and non-motor symptoms (*r_s_* = 0.33, *p* = 0.02) (Appendix A). The majority of patients reported improvements in dyskinesias, motor fluctuations, tremor, rigidity and side effects of medication, whereas gait, balance and non-motor symptoms were mostly reported as being worse. (Figure 2, Appendix A). Those who reported worsening in apathy (*r_s_* = 0.4, *p* = 0.03) and insomnia (*r_s_* = 0.43, *p* = 0.01) were less satisfied (Appendix A).

## 5. Discussion 

### 5.1. Rush Patient-Centered Outcomes Questionnaire (RUSH-DBS-Q)

To the best of our knowledge, this is the first study evaluating patients’ and caregivers’ satisfaction with DBS outcome in the long term (>5 years). Patient and caregiver perception is important given the elective nature of DBS surgery, especially in the context of a progressive neurological disorder. In this sample, the vast majority of patients did report satisfaction with surgery in the long term, despite their high pre-operative expectation target. While intuitively pre-operative expectations and satisfaction should be correlated, we did not find such relationship. Hasegawa et al. [6] did find a strong correlation between expectations and satisfaction in the short term but had a different methodological approach. In the future, we will adjust our questionnaire to better capture this relationship. The relationship between pre-operative expectations and post-operative satisfaction in this population remains unclear but the absence of a correlation does not negate the need to identify individualized expectations so that patients and caregivers can be adequately counseled regarding short- and long-term goals from DBS. 

### 5.2. Mobility 

Worsened mobility domain scores on the PDQ-39 correlated with lower satisfaction of DBS. Axial symptoms can be resistant to DBS in the long term and can have a significant impact on patient independence and QOL [9,10,11]. Maier et al. [12] found an inverse correlation between pre-operative axial scores and patient satisfaction with DBS at 3 months post-surgery. We did not find such association in our population possibly due to pre-operative screening methods at our institution. 

### 5.3. Non-Motor Symptoms 

Non-motor symptoms have a substantial impact on PD patients’ QOL [13]. The effect of DBS on non-motor symptoms has been evaluated at 24 months [14] but is not established in the long term. A worsening can have a negative impact on perceived benefit [15]. As discussed by Joint and Aziz [16], patients and caregivers may focus on the motor outcomes after surgery and fail to consider the influence non-motor symptoms may have on QOL. In this sample, a significant worsening in certain non-motor symptoms correlated with less satisfaction. In particular, apathy was associated with decreased patient satisfaction. It has been suggested that apathy post-STN DBS surgery could be directly related to the dramatic reduction in dopaminergic medications [17]; however, in our patient population, there was no correlation between change in LEDD and apathy. A likely explanation is that patients with a larger reduction in medication have a superior benefit of DBS or, alternatively, that patients having a higher dopaminergic medication have progressed at a relatively faster rate. There were no pre-operative predictors of post-operative apathy identified and pre-operative neuropsychological testing did not reveal a significant level of depression in any given patient. 

### 5.4. Study Limitations

There are limitations in our study worth noting. We used a locally developed, non-validated questionnaire to evaluate patient-centered outcomes. However, at the time the study was completed, no such instrument was available. In addition, the RUSH-DBS-Q was not completed pre-operatively, and all answers were obtained at the time of survey, more than eight years after surgery. We partially addressed partial recall by asking patients and caregivers to agree on answers to limit each individual’s recall bias. In addition, we did not repeat the UPDRS-III at the time of survey. This would have lent to a better understanding of the relationship between patient perception and motor outcome. Finally, all patients included in the study were under the care of one treating neurologist. Given the long follow-up since surgery, patients were likely satisfied with their care which could have led to bias when answering the satisfaction questions. However, to limit this bias, the treating neurologist was not involved in administration of the questionnaires. 

## 6. Conclusions

To our knowledge, this is the first study to assess long-term patient and caregiver satisfaction with STN DBS. After an average of eight years post-DBS surgery, the majority of patients were satisfied, felt they had made the correct decision to undergo DBS, would choose to have DBS again, would recommend DBS to others, and would have preferred to have surgery sooner. Pre-operative expectations were higher than expected based on our conservative pre-operative counseling, but they did not correlate with satisfaction. The latter should not negate the necessity of setting realistic expectations to caregivers and patients. We are currently collecting data prospectively to confirm the results of this pilot study.

## Figures and Tables

**Figure 1 brainsci-08-00060-f001:**
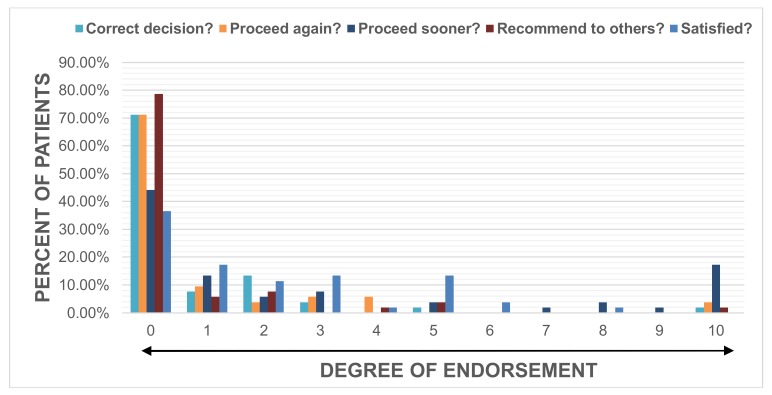
Patient endorsement. The percent of patients reporting degree of endorsement on the RUSH-DBS-Q. All questions were scored from 0 to 10, whereby 0 represented the most ‘positive’ answer. The questions centered around making the correct decision to undergo DBS, theoretical decision to proceed again with DBS, timing of DBS, confidence in recommending DBS to others, and overall satisfaction with DBS.

**Figure 2 brainsci-08-00060-f002:**
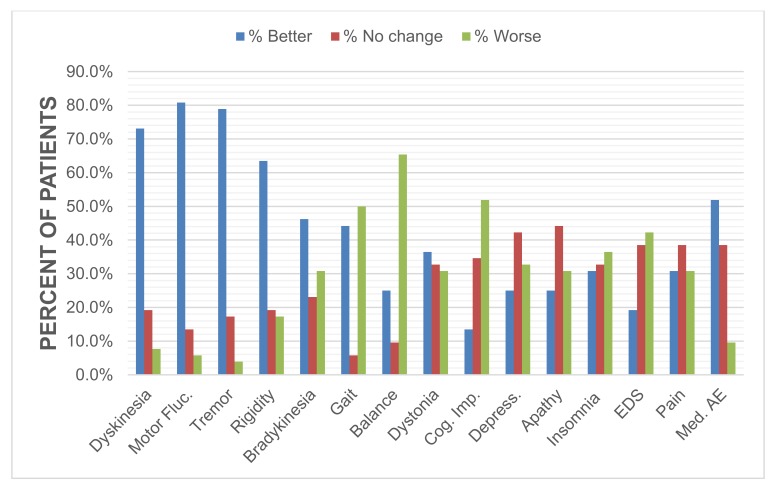
Change in self-rated symptom severity. All patients (*n* = 52) were asked to rate their PD motor and non-motor symptoms at two time points (pre-DBS and current time of 8.2 ± 2.6 years post-DBS). All items were rated based on severity, with 0 being the least severe and 10 being the most severe. Motor Fluc., motor fluctuations; Cog. Imp., cognitive impairment; EDS, excessive daytime sleepiness; Med AE, medication adverse effects.

**Table 1 brainsci-08-00060-t001:** Baseline Demographics and Pre-operative Data (*n* = 52).

Baseline Characteristic	Mean (SD) Score
Current age	66.1 (7.9)
Years since DBS surgery	8.2 (2.6)
PD duration at current time (years)	19.7 (5.6)
PD duration at time of surgery (years)	11.57 (4.7)
LEDD before surgery	1326 (652.5)
Pre-operative UPDRS-I	2.25 (1.78)
Pre-operative UPDRS-II, OFF	20.82 (5.47)
Pre-operative UPDRS-II, ON	8.27 (5.24)
Pre-operative UPDRS-III total, OFF, ON	43.38 (13.9), 16.86 (9.36)
Pre-operative UPDRS-III axial, OFF, ON	11.52 (4.51), 4.57 (2.73)
Pre-operative UPDRS-III rest tremor, OFF, ON *	3 (7), 0 (0)
Pre-operative UPDRS-III postural tremor, OFF, ON *	1.5 (2.5), 0 (1)
Pre-operative UPDRS-III bradykinesia left OFF, ON	7.71 (2.94), 3.27 (2.27)
Pre-operative UPDRS-III bradykinesia right OFF, ON	6.69 (2.46), 2.52 (1.7)
Pre-operative UPDRS-III rigidity OFF, ON	9.4 (4.33), 3.99 (3.65)
Pre-operative UPDRS-IV	8.36 (3.25)
Pre-operative S&E-OFF	58% (20%)
Pre-operative S&E-ON	85% (10%)
Pre-operative PDQ-39 Summary Index	33.01 (12.1)
Pre-operative PDQ-39, Mobility domain	46.25 (20.4)
Pre-operative PDQ-39, ADLs domain	44.53 (16.16)
Pre-operative PDQ-39, Emotional well-being domain	28.65 (18.51)
Pre-operative PDQ-39, Stigma domain	32.68 (22.8)
Pre-operative PDQ-39, Social support domain	14.92 (17.92)
Pre-operative PDQ-39, Cognitive impairment domain	23.52 (15.1)
Pre-operative PDQ-39, Communication domain	26.39 (19.93)
Pre-operative PDQ-39, Bodily discomfort domain	47.57 (25.46)

Values with * represent medians (Interquartile range).

**Table 2 brainsci-08-00060-t002:** Change in Scores from Pre-DBS to Current Time (8.2 ± 2.6 years post-DBS).

Outcome Measure	All Patients (*n* = 52)	*p*-Value
LEDD	−634.8 (721.4)	<0.0001
UPDRS-I *	1 (2)	0.0002
UPDRS-II, OFF	−0.59 (9.3)	0.66
UPDRS-II, ON	7.9 (8)	<0.0001
UPDRS-IV	−4.6 (3.4)	<0.0001
S&E-OFF	−2.0% (32.0%)	0.71
S&E-ON	−17.0% (23.0%)	<0.0001
PDQ-39, Summary Index	2.1 (18.2)	0.44
PDQ-39, Mobility domain	9.7 (31)	0.03
PDQ-39, ADLs domain	−6.2 (27.4)	0.13
PDQ-39, Emotional well-being domain	1 (24.7)	0.77
PDQ-39, Stigma domain	−10.1 (22.2)	0.003
PDQ-39, Social support domain	6.3 (26.9)	0.11
PDQ-39, Cognitive impairment domain	4.6 (22.8)	0.16
PDQ-39, Communication domain	22.4 (26.2)	<0.0001
PDQ-39, Bodily discomfort domain	−13.2 (30.1)	0.004

Values represent means and SD. Values with * represent medians and IQR. For the UPDRS I, II, IV, and PDQ-39 higher scores indicate worse severity. A positive number indicates a worsened score from baseline, whereas a negative number indicates an improvement in score from baseline. For S&E lower scores indicate worse severity. A positive number indicates an improvement from baseline, whereas a negative number indicates a worsening from baseline.

**Table 3 brainsci-08-00060-t003:** Correlation between Change in Pre-DBS Scales and Satisfaction.

Outcome Measure	Spearman Correlation Coefficient	*p*-Value
LEDD	0.25	0.09
UPDRS-I	0.44	0.002
UPDRS-II, OFF	0.32	0.02
UPDRS-II, ON	0.21	0.14
Baseline (UPDRS-III-OFF–UPDRS-III-ON) *	−0.01	0.93
UPDRS-IV	0.34	0.02
S&E-OFF	−0.44	0.002
S&E-ON	−0.34	0.02
PDQ-39, Summary Index	0.57	<0.0001
PDQ-39, Mobility domain	0.57	<0.0001
PDQ-39, ADLs domain	0.31	0.03
PDQ-39, Emotional well-being domain	0.40	0.005
PDQ-39, Stigma domain	0.40	0.005
PDQ-39, Social support domain	0.45	0.001
PDQ-39, Cognitive impairment domain	0.38	0.01
PDQ-39, Communication domain	0.36	0.01
PDQ-39, Bodily discomfort domain	0.32	0.02

Values represent correlation of current satisfaction with change in scales from pre-DBS to current time (8.2 ± 2.6 years). * Represents pre-operative levodopa response, UPDRS-III-OFF–UPDRS-III-ON.

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
