# Peer review of "Long-Term Satisfaction and Patient-Centered Outcomes of Deep Brain Stimulation in Parkinson’s Disease"

_brainsci, 2018, doi:10.3390/brainsci8040060_

Round 1

Reviewer 1 Report

In this revised manuscript, Karl et al have re-worked their analyses to more accurately reflect the correlations and conclusions that can be drawn from a single measurement of the RUSH-DBS-Q about patient perceptions of their DBS therapy. This version presents a much cleaner and succinct analysis. I would suggest modifying the title to reflect the "satisfaction" assessment, which is otherwise under-represented in the literature and is the main point made in their conclusion. This may draw more attention to their work (e.g., "Long term satisfaction and patient centered outcomes...."). I would also suggest combining tables 1 and 2 for clarity so the reader can more easily view the differences in pre and post DBS assessments.  The lack of objective UPDRS III ratings at the time of collection of patient reported outcomes remains a limitation that should be acknowledged; having it would lend the reader a sense of the correlation between patient perceptions as reported in ESuppfile4 and objective measurements, especially since a pre-op UPDRS is reported.

Author Response

Comment 1: I would suggest modifying the title to reflect the "satisfaction" assessment, which is otherwise under-represented in the literature and is the main point made in their conclusion. This may draw more attention to their work (e.g., "Long term satisfaction and patient centered outcomes....").

This has been changed.

Comment 2: I would also suggest combining tables 1 and 2 for clarity so the reader can more easily view the differences in pre and post DBS assessments. 

Thank you for your recommendation. We prefer to keep the demographic table separate from the other tables. Since Table 2 includes the change in scores from pre-DBS to current time it does not have several values that are depicted in Table 1 (UPDRS III, age, PD duration, etc.).

Comment 3: The lack of objective UPDRS III ratings at the time of collection of patient reported outcomes remains a limitation that should be acknowledged; having it would lend the reader a sense of the correlation between patient perceptions as reported in ESuppfile4 and objective measurements, especially since a pre-op UPDRS is reported.

This is now discussed in the limitation section (lines 188-190).

Reviewer 2 Report

I am happy with the changes that have been made

Author Response

The reviewer has not requested further changes. We have updated the limitation section discussing the lack of objective UPDRS-III ratings and how this would have lent to a better sense of the correlation between patient perception and objective measurements.